# Sex and Gender Differences in Neurodegenerative Diseases: Challenges for Therapeutic Opportunities

**DOI:** 10.3390/ijms24076354

**Published:** 2023-03-28

**Authors:** Annalisa Bianco, Ylenia Antonacci, Maria Liguori

**Affiliations:** National Research Council (CNR), Institute of Biomedical Technologies, Bari Unit, 70125 Bari, Italy

**Keywords:** neurodegenerative diseases, autoimmunity, sex/gender differences, disease heterogeneity, multiple sclerosis, Parkinson’s disease, Alzheimer’s disease, amyotrophic lateral sclerosis, spinal muscular atrophy

## Abstract

The term “neurodegenerative diseases” (NDs) identifies a group of heterogeneous diseases characterized by progressive loss of selectively vulnerable populations of neurons, which progressively deteriorates over time, leading to neuronal dysfunction. Protein aggregation and neuronal loss have been considered the most characteristic hallmarks of NDs, but growing evidence confirms that significant dysregulation of innate immune pathways plays a crucial role as well. NDs vary from multiple sclerosis, in which the autoimmune inflammatory component is predominant, to more “classical” NDs, such as Parkinson’s disease, Alzheimer’s disease, amyotrophic lateral sclerosis, and spinal muscular atrophy. Of interest, many of the clinical differences reported in NDs seem to be closely linked to sex, which may be justified by the significant changes in immune mechanisms between affected females and males. In this review, we examined some of the most studied NDs by looking at their pathogenic and phenotypical features to highlight sex-related discrepancies, if any, with particular interest in the individuals’ responses to treatment. We believe that pointing out these differences in clinical practice may help achieve more successful precision and personalized care.

## 1. Introduction

Despite the skepticism of the past, growing evidence suggests that inflammation does affect the central nervous system (CNS) also during nonautoimmune neurodegenerative diseases (NDs). Widespread inflammation within the brain, mostly mediated by microglia and astrocytes, seems to anticipate the deposition of misfolded proteins characteristic of NDs like prion disease and Alzheimer’s disease (AD) [1,2]. Furthermore, the evidence of natural antibodies (nAbs) highly specific against, e.g., neuronal tissue, beta-amyloids, and the 200 kDa heavy neurofilament subunit (NH-F) has been reported in AD [3], thus confirming the hypothesis that there may be an overlapping between the autoimmune and the NDs [4].

On this ground, an ever-increasing group of neurological diseases have become appreciated as encompassing a significant immunological component in their pathogenesis, which may explain, at least partially, significant heterogeneity in clinical phenotypes and treatment response. This has been described in pathologies like multiple sclerosis (MS), in which the autoimmune inflammatory component is predominant but neurodegeneration plays a key, although controversial, role [5,6], as in more definite NDs, such as Parkinson’s disease (PD), AD, amyotrophic lateral sclerosis (ALS), and spinal muscular atrophy (SMA). 

Of interest, most of the clinical differences reported in NDs seem to be closely linked to sex, which may be justified by female/male significant changes in autoimmune mechanisms. It is worthy to note that microglia showed sexually dimorphic roles in the developing brain, which might explain some sex-differences in NDs [7]. 

However, what is often overlooked is that a very significant component of the genetic contribution to autoimmunity is gender, with the presence or absence of a Y chromosome influencing the risk for autoimmune disease [8]. Even if the term “gender” refers to the differences between males and females in cultural context while “sex” refers to the chromosomal set, XX or XY, in this review, the general use of both terms refers to a complex biological state that includes genetic, environmental, and epigenetic factors (Figure 1).

This review aims to highlight the sex/gender variations in NDs; far from considering all the pathologies covered by this term, we decided to analyze a list of conditions that, in our opinion, were representative of a peculiar group of disorders. Starting from a complex multifactorial autoimmune disease (MS), moving on to the “more common” PD, AD, and ALS, and finally to a genetically-based disease (SMA), by looking at their pathogenic and clinical features, with a particular interest in the response to treatment. We believe that pointing out these differences in clinical practice may help in understanding the efficacy of a given treatment, thus leading to more successful precision and personalized care. 

## 2. Multiple Sclerosis

MS is a chronic autoimmune inflammatory and demyelinating disease of the CNS that causes physical disorders up to disability [9]. It mainly affects young people, roughly 90% between the ages of 15 and 50 [10]. From the immunological point of view, MS was found to be characterized by a reduced self-tolerance towards the CNS myelin proteins [11].

MS is a heterogeneous disease. Clinically, most MS patients (70%) show at the onset a disease course named relapsing-remitting (RR), characterized by episodes of clinical/MRI activity followed by nearly complete functional recovery. Over time, some of them (20–30%) complain of a progressive accumulation of disability, possibly due to concurrent neurodegenerative processes; they are so-called secondary progressive MS (SPMS), also different from the remaining primary progressives (PP, around 30%), who suffer from a slow and progressive accumulation of disability since the disease onset [12]. 

The MS pathogenesis is still under discussion, although it is known that there are confirmed risk factors, such as the presence of the allele HLA-*DRB1*1501* in the MHC complex of affected subjects and environmental factors like vitamin D level, obesity, smoking, and the Epstein-Barr virus (EBV) infection [13].

Sex is another of the top risk factors in MS, with females being more often affected than males (3:1 ratio), as well as sex differences in clinical courses, disease severity, and progression having been reported. Several pieces of evidence showed that in fact females have a higher cumulative hazard risk of relapses than males, who are more likely to develop PPMS with a slowly worsening myelopathy, progressive weakness, and abnormal gait [10]. Furthermore, males with RRMS accumulate disability and progress to SPMS significantly faster than females, as well as show worse cognitive impairment and gray matter atrophy compared with females. Of interest, despite the female prevalence, the overall disability progression is comparable between the two sexes [14]. A potentially higher inflammatory activity in female patients with MS is supported by radiological evidence, indicating that they have more contrast-enhancing lesions compared to male patients [15]. Above all, it seems that males show more degenerative processes associated with gray matter pathology and atrophy [16].

Several hypotheses can be made to explain these sex discrepancies, but it is reasonable to believe that immunological mechanisms may drive the sex and gender bias, driven by hormonal as well as genetic and epigenetic factors [15]. As an example, the innate immune response differs between males and females: males present a higher number of natural killer cells compared to females, while phagocytic activity of neutrophils and macrophages is higher in females, and the antigen-presenting cells (APCs) seem to be more competent than males (Table 1) [11].

Sex hormones (mainly androgens and estrogens) can influence the immune system and its response through their own receptors, possibly modifying the risk of MS as well as some of its clinical features. In female patients, hormonal fluctuations during the menstrual cycle may affect MS relapses, whereas women with MS experience fewer relapses during their pregnancies and an increased number in the post-partum period. A possible explanation can be increased levels of immunosuppressive interleukin-10 (IL-10) in the third trimester of pregnancy and a T-lymphocyte helper 2 (Th2) skew in cytokine production and responses. Post-partum relapses may be associated with increased IL-8 production [19].

In MS therapy, the main goal is to at least slow down the disease, e.g., by reducing the inflammatory cascades and the myelin injury [13]. At present, the available long-term treatments, known as disease-modifying therapies (DMTs), can be grouped into three different classes:
1-Early traditional injectable DMTs, such as interferon beta (IFNβ-1a and -1b), that seem to act by suppressing T cell activity, inducing IL-10 production, and altering the differentiation of CD4^+^ T cells toward a Th2 phenotype [13]; and glatiramer acetate (GA), that targets myelin-specific autoantibodies in order to reduce autoreactivity and promote a predominant Th2 phenotype [13]. 2-Oral medications, represented by sphingosine 1 phosphate (s1p) receptor modulators such as Fingolimod, alter the immune migration by binding S1P receptors on lymphocytes, inducing sequestration of circulating mature lymphocytes [9], and Teriflunomide, which acts by effecting rapidly dividing lymphocytes [13]. More recently, Cladribine is a nucleoside analogue that inhibits DNA synthesis and DNA chain termination with cytotoxic activity towards lymphocytes and monocytes [9]. Among the fumaric acid derivates, dimethyl fumarate (DMF) suppresses Th1/Th17 inflammatory responses and promotes a Th2-type response. It is also reported that DMF reduces IL-17 production by CD4 T cells [20].3-Infusion and injectable DMTs are a group of monoclonal antibodies that are comprised of Natalizumab, which prevents lymphocyte adhesion and migration from the peripheral vascular bed to the CNS; Alemtuzumab, which selectively binds the CD52 protein, inducing the clearance of T and B cells and increasing the secretion of brain-derived neurotrophic factor (BDNF) [9,13]; Rituximab, which targets CD20; and Ocrelizumab, which induces an antibody-dependent cytolysis and complement-mediated lysis of B cells [9]. Recently, Ofatumumab, another monoclonal antibody that also binds to CD20^+^ B cells, resulting in B cell depletion, has been approved and is currently administered to MS patients [13].

In MS, few studies have been focused on the evaluation of drug responses by sex so far. An analysis of the IFNβ therapy efficacy in RRMS patients revealed that during their reproductive years, females showed a more robust immune response, a higher absolute number of CD4^+^ lymphocytes, and a higher production of Th1-derived cytokines than males (e.g., interferon gamma, IFNγ, and IL-12), that might explain the different response to the same drug [39]. These different results seem to be dependent on the age of treatment onset and the number of pre-treatment MS attacks. In contrast, an analysis of outcomes in a cohort of SPMS patients revealed that females treated with IFNβ progressed more slowly than males [19]. Furthermore, a recent report on adult MS patients treated with Fingolimod demonstrated that mostly female patients complained of an infectious episode after the first year of therapy [40], thus confirming that there may also be sex-based differences in the occurrence of side-effects.

## 3. Parkinson’s Disease

PD is a neurodegenerative multi-systemic alpha-synucleinopathy. The aggregation and accumulation of α-synuclein (SNCA), a presynaptic neuronal protein, in Lewy bodies and neuritis within the CNS is in fact considered a pathological hallmark of PD [41,42,43].

In 2019, global estimates showed over 8.5 million individuals with PD (Fons: WHO. Parkinson Disease. https://www.who.int/news-room/fact-sheets/detail/parkinson-disease (accessed on 27 March 2023)). Epidemiological evaluations confirmed an age-related incidence that increases 5–10 fold from the sixth to the ninth decades of life. Most PD cases are sporadic with multifactorial origins, whereas familial cases are caused by genetic factors. Mutations in more than 30 loci, such as parkin (*PRKN*), synuclein (*SNCA*) and leucine-rich repeat kinase 2 (*LRRK2*), have been associated with PD [44]. The diagnosis is mostly based on clinical criteria, including history and physical examination [45]. The clinical pattern of PD is characterized by motor and nonmotor symptoms [45], with the death of dopaminergic neurons in the substantia nigra pars compacta (SNpc) being most likely the pathogenic justification of motor symptoms [46]. 

For the treatment of motor symptoms, levodopa (LV) is usually combined with a peripheral decarboxylase inhibitor, synthetic dopamine receptor agonists, centrally acting antimuscarinic drugs, amantadine, monoamine oxidase-B (MAO-B) inhibitors, and catechol-O-methyltransferase (COMT) inhibitors. Adjunctive therapy with COMT and MAO-B inhibitors or dopamine agonists is necessary to manage motor fluctuations [47].

Recent evidence underlines various immune alterations in PD. Aggregates of α-synuclein trigger the activation of microglia through the presentation of the SNCA epitopes in the context of APCs and MHC molecules, thus altering the neurotransmitter pathways [48]. Along with the T cell responses, the active B cells are able to produce autoantibodies against the antigen SNCA [49], monosialotetrahexosylganglioside GM1-gangliosides, catecholamine-based melanins, and others [48]. Activated microglia elicited the expression of pro-inflammatory cytokines. A significant increase in innate immune factors, including IL-1, IL-2, IL-6, and tumor necrosis factor (TNF)-α in the SNpc, in the peripheral blood, and in the cerebrospinal fluid (CSF) of PD patients has also been reported (Table 1) [50]. 

Genetic, epigenetic, and environmental factors seem to be involved in the sex-related differences in PD risk and clinical courses. Increasing evidence does show that the disease affects women and men differently. The risk of developing PD is 1.5 times greater in men than in women, possibly due to estrogens, although women have a higher mortality rate and faster progression of the disease. However, men have an earlier onset of PD [8]. Distinctive symptoms are reported in women, such as differences in the response to pharmacological therapies and the deep brain stimulation procedure [51]. Many studies reported sex-based differences in motor and non-motor symptom patterns. In women, motor symptoms emerge later with specific traits such as reduced rigidity, tremor, a higher propensity to develop postural instability, and an elevated risk for motor complications related to LV treatment [52]. PD males were associated with later development of freezing of gait and a higher risk for developing camptocormia [53,54]. Non-motor symptoms, such as depression, restless legs, constipation, loss of taste or smell, pain, and excessive sweating [55], are more severe and common in women [52]. Conversely, men exhibit greater sexual dysfunction and deterioration of their sexual relationship than women [56]. Additionally, a sex-specific pattern for cognitive changes associated with Parkinson’s disease was found. Deficits in verbal fluency and recognition of facial emotions are prevalent in men, while a reduction in visuospatial cognition is more likely in women [57,58,59]. 

Biological differences by sex in the nigrostriatal dopaminergic system seem to play an important role in the vulnerability of the complex molecular network implicated in PD. In men, SNpc is in fact characterized by a higher number of dopaminergic neurons, increased expression of genes involved in PD like *SNCA* and *PINK1*, more significant dopamine release induced by stimuli, and increased vulnerability to drugs. On the other hand, genes implicated in signal transduction and neuronal maturation are upregulated in females, as dopaminergic cells show lesser vulnerability to degeneration and other factors than men [52,60]. Finally, the impact of estradiol in the synthesis, release, reuptake, and turnover of dopamine has been demonstrated in animal models of PD [52], thus explaining the potential protective effect of estrogens in dopaminergic neurons with possible inhibitory effects on the formation and stabilization of α-synuclein fibrils [61]. Sex differences have been reported in pharmacokinetics and pharmacodynamics mechanisms during the administration of PD drugs. Women not only presented a greater bioavailability of LV and lower LV clearance levels, but they were also known to develop LV-induced dyskinesias more frequently and had an increased risk of the “brittle response” to LV [62]. A recent study on LV in naive PD patients showed that women seemed more prone to develop LV-related complications and exhibited a higher LV bioavailability compared with men [63]. Furthermore, women showed a slower gastric emptying time than men, with approximately 25% less COMT enzyme activity, and their transit time variability was critical for intestinal regional differences in absorption, such as LV [63,64]. This evidence may partially explain the pharmacokinetic differences and LV effects related to gender. Other factors, including body weight, abnormal plastic responses to LV, and differences in energy metabolism, may all contribute to the gender differences in LV complications [65]. 

Different genotypes may influence the therapeutic response of LV related to sex. Carriers of *MAO-B* (rs1799836) A and AA genotypes and *COMT* (rs4680) LL genotype suffered more frequently from LV-induced dyskinesia, and male individuals carrying the *MAO-B* G allele showed an increased risk of 2.84-fold to develop motor complications when treated with higher doses of LV. The *MAO-B* encoding gene is located on chromosome X, supporting the hypothesis of different dopamine metabolism between men and women [66].

Women seem to be more susceptible to the gastrointestinal and orthostatic adverse effects of Tolcapone, a COMT inhibitor used as an adjuvant for LV. A higher bioavailability of Pramipexole, a dopaminergic agonist, was observed in women, probably related to a lower oral clearance [67]. A higher prevalence of dyskinesia in women at baseline after treatment with safinamide was also reported. However, a similar reduction of patients with dyskinesia in both genders over a 1-year follow-up was observed, and the prevalence of any fluctuations was similarly reduced in both females and males after safinamide introduction [68].

Among the PD treatments, deep brain stimulation of the subthalamic nucleus (STN DBS) is the mainstream surgical procedure for advanced PD, and it is becoming more popular nowadays as a treatment. Clinical studies showed that after STN DBS, women had a greater improvement in activities of daily living than men and a positive effect on mobility and cognition, though bradykinesia was less responsive to STN DBS in women than in men [69]. Although there are some studies about the impact of gender on medical LV treatment in PD, poor evidence is still available on other drugs. More studies are needed to better understand how gender differences can alter drug responses in PD.

## 4. Alzheimer’s Disease

Dementia is a syndrome that leads to progressive deterioration of cognitive functions, impacting each patient’s social and occupational abilities. The term “dementia” encompasses different conditions including AD, vascular dementia, frontotemporal dementia, and others. AD is the most common form of dementia, comprising 60–70% of all cases (Fons WHO. Dementia. https://www.who.int/news-room/fact-sheets/detail/dementia (accessed on 27 March 2023)). The hallmark features of AD include the presence of extracellular amyloid-β plaques (Aβ), intracellular neurofibrillary tangles, and neurodegeneration. Of interest, AD is described as a brain-centric disorder of innate immunity involving simultaneous autoimmune and autoinflammatory mechanisms [70]. In response to initial (yet unknown) stimuli, Aβ is released as an early reactive immunopeptide, exhibiting both immunomodulatory and antimicrobial properties (in the presence or absence of bacteria), and triggering an innate immune cascade that causes a misdirected attack against “self” neurons. As a result, the produced necrotic neuronal degradation spread to adjacent neurons, causing further Aβ release, which leads to a self-perpetuating chronic autoimmune cycle [71]. The microglial activation is correlated with amyloid deposition in AD, which is connected to the stage of the disease and the affected brain region [72]. Amyloid and microglia interact with each other in AD progression in a sex-dependent manner, as demonstrated in a different mouse model in which the pathological processes were found more severe in females (also in Table 1) [73,74].

At present, pharmacologic approaches provide modest symptomatic relief, as they include inhibitors of the acetylcholinesterase enzyme (AChE) and N-methyl-D-aspartate (NMDA) receptor antagonists. However, immunotherapies in AD have been the most studied strategies in recent years [75]. 

Epidemiological data show that the prevalence of AD differs between men and women, the latter having a higher risk and incidence of developing AD than men (approximately 2:1), with more progressive cognitive and physical decline [25]. Women and men also exhibit different clinical features, in terms of disease duration (longer in women) and concomitant neuropsychiatric disturbances. Women affected by AD experience a faster progression of hippocampal atrophy and show a greater load of AD pathology (i.e., amyloid plaques and neurofibrillary tangles) [76]. In addition, the amyloid-β load and tau deposits detected by positron emission tomography (PET) appeared earlier in women than in men in individuals at risk of developing AD [77]. Differences in life expectancy, educational level, cognitive detection biases, sex hormones, and genetics could all explain such imbalances. 

The functional role of sex hormones such as estrogens, progesterone, and androgens in brain development as well as in aging and AD processes has been recognized. The age-related drastic loss of sexual hormones, resulting in the depletion of post-menopausal estrogens, may explain the higher prevalence of AD in women [78]. In men, the loss of steroid hormones was less drastic. Even if the level of circulating testosterone exhibited a gradual decline over time, its reduced levels due to aging may also increase the occurrence of the disease in men [25]. Moreover, the reduction in estradiol levels during and beyond the fifth decade of life may be responsible for deficits of cerebral metabolism and vascular pathologies mainly among women, since males of the same age would continue to aromatize testosterone into estrogens [79]. Changes in sex hormone receptors and downstream signaling pathways have also been found during aging. Indeed, there is an increased expression of non-functional splice variants of estrogen receptor alpha in the hippocampus, with higher expression levels in elderly females than in males [80]. In the sporadic forms of AD, the association of homozygous single nucleotide polymorphisms (SNP) of *ESR1* and *ESR2* with apolipoprotein E4 (*ApoE4*) conferred an increased risk of cognitive impairment in both sexes, with a higher prevalence in women. Indeed, the research data points out that the risk of AD is even more pronounced in women carriers of the e4 allele of *APOE* than men [25]. 

Apart from hormones, sex chromosomes contribute to the heterogeneity of AD. X chromosome aneuploidy may contribute to processes that lead to pathological changes in AD and aging brains. On the other hand, an extra X chromosome could explain some protective effects against AD, possibly by increasing the expression of genes that evade X inactivation, as in the case of the *PCDH11X* gene [80]. An interesting study found a SNP in the *PCDH11X* gene associated with higher risks of developing AD in women [81]. Moreover, the loss of chromosome Y (LOY) is the most common acquired mutation in aged men that increases the risk of developing AD [82]. Even if the role of genes in AD requires further study, many reports have noted a marked expansion of genes between males and females, including variants within the genes *OSTN*, *CLDN16*, *KDM6A*, *MGMT*, *SERPINA1*, *USP11*, and *ST2* [83,84,85]. Recently, Devis et al. found an increased expression level of 19 genes on the X chromosome associated with slower cognitive decline only in women, suggesting a possible protection from cognitive decline in aging and AD [86]. Interestingly, the genes *TREM2*, *GRN*, and *IGF-2* are considered to be male-specific risk genes for AD [80]. Evidence also underlines the impact of sex on the epigenomic signatures of AD; hypomethylation of CpG islands in the promoter region of the *AURKC* gene has been reported in male AD patients, whereas hypermethylation has been reported in female AD patients [87].

Reported differences in the activation of the immune system may be a causal factor in the dimorphism of AD. Females exhibit a stronger immune response than males with increased inflammatory cytokines, chemokines, and gliosis [11,26]. The differential effects of pro-inflammatory events upon microglia in males and females during development suggests that in developing males, the microglia may be more sensitive to inflammatory events. There is also evidence that differential responses of microglia exist in adulthood depending on sex. Even if microglia in the adult male brain appear to be more reactive to inflammatory events compared to females, inflammatory events in adulthood appear to promote age-related microglial dysfunction in females. However, microglia shift to a more pro-inflammatory state with reduced homeostatic functions during aging in both sexes [7]. Sexually dimorphic immune responses could also arise due to incomplete inactivation of the X chromosome in females. Many genes located on the X chromosome have been linked to immune-related functions, and around 23% of X-linked genes retain augmented expression in females [88].

At present, there is no effective cure for AD, but there are ongoing trials on treatments that seem to modify the progression of the disease, as well as pharmacological and non-drug options for treating some symptoms. Gender differences seem to influence the outcome of psychosocial interventions. Women were shown to significantly improve with psychosocial interventions in relation to behavioral and psychological problems. On the other hand, in studies investigating the impact of music therapy, being male is associated with worse outcomes in terms of increased physical aggression [89]. Furthermore, gender may influence the response to acetyl cholinesterase inhibitors, as women appear to be more likely to respond to Donepezil and Rivastigmine. However, sedative-hypnotics are overprescribed to women, which leads to an increased risk of side effects. In particular, bradycardia caused by cholinesterase inhibitors affects more women, while men suffer more emergency hospitalizations and deaths after the prescription of antipsychotics [90]. However, we need more clinical and preclinical studies that indicate sex as a biological variable to be considered in the evaluation of the disease itself or in the response to pharmacological interventions.

## 5. Amyotrophic Lateral Sclerosis

ALS is a disease characterized by degeneration of both upper and lower motor neurons in the brain and spinal cord [32]. It shares pathological aspects with frontotemporal dementia (FTD), and in fact, patients can show features of both disorders in 5–15% of cases [30]. ALS can onset with spinal or bulbar symptoms [32]; unfortunately, the progression of most of the courses is rapid, so patients usually die within 3–4 years from the onset, mostly because of respiratory failure. 

Most ALS cases are sporadic (sALS), with 7% of them associated with mutations in more than 20 genes, among them *C9orf72* (encoding guanine nucleotide exchange C9orf72), *TARDBP*, *SOD1*, and *FUS* [30]. A family history (fALS) is found in a small percentage of patients (10%). Recently, other genes have been associated with ALS (*MATR3*, *CHCHD10*, *TBK1*, *TUBA4A*, *NEK1*, *C21orf2*, and *CCNF*); they are involved in different molecular pathways, with mutations resulting in defects in the protein clearance pathway, mitochondrial dysfunction, altered RNA metabolism, impaired cytoskeletal integrity, altered axonal transport dynamics, and DNA damage accumulation due to defective DNA repair [91].

Evidence also showed that the main neurodegenerative process underlying ALS is characterized by an autoimmune signature. As confirmation, the degree of T-lymphocytic infiltration observed in the anterior horn of the spinal cord suggests that T-cells and macrophages may contribute to spinal cord and brain inflammatory mechanisms [34]. Furthermore, in ALS, a crucial role seems to be played by IL-17A: an IL-17A-mediated pathway seems to in fact activate the local glial cell, which is a crucial step for the neurodegenerative process. As confirmation, patients with ALS showed increased serum and CSF levels of this cytokine (Table 1) [92].

ALS is more common in men than in women, with a male/female ratio of 1.2–1.5:1; their life risk is about 1:350 for men and 1:400 for women [30]. This sex-related differences have deeply been studied for age and site of onset, prognosis, and cognitive profile, considering the significant impact of these features in disease heterogeneity. Men have a more likely younger onset of ALS; according to the site of onset, a greater proportion of women complain of bulbar onset than do men, who more often presented with spinal/limb onset [93]. In patients with younger onset, the upper-limb involvement is predominant in men than women [93]. When cognitive functions were impaired, females were reported having a double risk for a greater executive impairment than males [94]. In neuroradiological evaluations, females showed a higher cortical thickness in the right parieto-occipital and left mid-frontal regions, whereas males demonstrated higher cortical thickness in the left lingual and left superior temporal regions; other results indicate significant gender differences in the frontotemporal and cerebellar regions [95].

In ALS, gender-related pathological and phenotypical differences may also be influenced by hormones. It is known that in females, sex hormones not only change during the menstrual cycle but also differ in the expression of their receptors (estrogen receptors- ERs and progesterone receptors-PRs). It seems that androgens might affect motoneuron development and axonal regeneration after injury in both sexes [60]. In addition, while in female controls the circulating levels of sex hormones significantly declined with age, testosterone levels remained elevated with increasing age in ALS patients; notably, in ALS, higher testosterone levels and a lower progesterone/free testosterone ratio correlated with a faster worsening of respiratory parameters [60]. In a recent study on the SOD1 ALS mouse model, progesterone was found to slow down the progression of the disease and extend the life span of the affected male mice without delaying the symptom onset [96]. The protective action of female sex steroids may be due to their ability to prevent cell death with a direct action on the receptors expressed by the motor neuron (ERs) and muscle cells and decrease the inflammatory component of the disease; it was also demonstrated that treatments with 17β-estradiol delayed the disease progression in ALS mice [60].

Sexual differences have been also observed in the levels of circulating markers. A significant overexpression of many microRNAs (miRNAs) was found in male patients with ALS vs. females. Among others, a group of muscle-specific “myo-miRNA”, (which include miRNA-206, miRNA-155, miRNA-221, and miRNA-146) that not only seem to be involved in the regulatory and functional response to cell damage but also in the immune response, negatively regulating the secretion of pro-inflammatory cytokines, thus preventing aggressive inflammation. Therefore, miRNAs may play a role in the pathogenesis of ALS due to their link between muscle atrophy and inflammation through the action of nuclear factor kappa-light-chain-enhancer of activated B cells (NF-kB). Differential miRNA expression in ALS muscle suggests a different disease course related to gender due to differences in hormonal control [97]. 

Unfortunately, ALS still lacks efficient therapy, so at present, its management is essentially focused on alleviating symptoms and improving the quality of life and survival of patients with a multidisciplinary approach. The only available long-term drug is RILUZOLE (2-amino-6-trifluoromethoxy benzothiazole, RP 54274, Rilutek™), designed for slowing the progression of the disease. Although there is the potential for improving neuronal activity, at present the results are quite disappointing [98]. Up to date, no data have been published on sex-related responses to treatment.

## 6. Spinal Muscular Atrophy

SMA is a neuromuscular disease characterized by the loss of motor neurons and progressive muscle wasting. Based on the age at onset and the severity of the symptoms, there are multiple forms of SMA, including acute infantile (SMA type I), chronic infantile (SMA type II), chronic juvenile (SMA type III), and adult (SMA type IV). The gene involved in the disease is *SMN1*, which was identified in the SMA locus on chromosome 5q. *SMN1* is homozygously deleted in 95% of SMA patients and deleteriously mutated in the remaining patients [36]. *SMN2* is a paralog of SMN1, which differs by a single nucleotide substitution in exon 7, determining a splicing defect with the exclusion of exon 7 from *SMN2* mRNA. This change produces a truncated and unstable protein in around 90% of cases. *SMN2* can be present in multiple copies and acts as a genetic modifier of disease severity [99].

Currently, therapies in use include Nusinersen/Spinraza™ an antisense oligonucleotide therapy; Onasemnogene abeparvovec/Zolgensma™ as an AAV9-based gene therapy; and Risdiplam/Evrysdi™ as a small molecule modifier of pre-mRNA splicing [100]. Although both genders benefit from the effectiveness of these drugs, no sex-specific differences in therapeutic response and side effects have been reported in preclinical and clinical studies.

Several studies have highlighted the role of immune activation in NDs, including motor neuron disorders [37,101], but at present, the role of the immune system and inflammation in the pathogenesis of SMA and its importance as a therapeutic target are still unknown. In 33 pediatric and adult SMA patients, an inflammatory cytokine signature was found in both serum and CSF as an index of inflammation and immune system activation at peripheral and central levels [102]. Furthermore, peripheral immune organ abnormalities and T-cell maturation dysfunction have emerged in mouse models of SMA, and it is reasonable to assume that these intrinsic T-cell alterations may result in an abnormal neuroinflammatory response and disease exacerbation [7]. Increased astrogliosis was observed in necropsies of patients [103] and also in the mouse model with selective deletion of exon 7 in the *SMN* gene (SmnΔ7) both at pre-symptomatic and symptomatic stages [104], whereas microglial activation was only seen in the SmnΔ7 mouse model but not in the more severe mice with the homozygous Smn knockout allele (Smn −/−; SMN2) (Table 1) [105,106].

Healthy males and females display differences in circulating hormones, mitochondrial content, biogenesis, and activity, as well as muscle physiology, including muscle size, muscle fiber composition, and anabolic and catabolic factors [36]. These physiological gender-related differences may impact the different clinical outcomes and disease progression and severity during SMA. Epidemiologically, more males were affected by mild SMA than females [107]; in particular, the onset of mild SMA in females seems delayed by about three years as compared to males [107]. While a Japanese study on 122 patients with SMA (60 females and 62 males) revealed a predominance of male patients with SMA type 3 (the walker group) without deletion within the *NAIP* gene or with high *SMN2* copy number (3 or 4 copies), no significant gender differences were found in the number of patients with SMA type 1 or 2 [107]. However, motor function impairment seemed to occur more predominantly in male patients. Even if the median Hammersmith Functional Rating Scale Expanded (HFRSE) score were significantly lower in males than in females (16 vs. 40), the upper limb segment of the battery did not reveal a statistically significant difference between the sexes [99]. Of interest, the presence of gender differences in the growth pattern of SMA children has been described. Girls showed a more linear weight and supine length growth compared to boys, with SMA1 males having a decrease in weight, length, and BMI velocity, whereas a small proportion of SMA2 males presented a higher body weight than the general pediatric population [108]. 

Mitochondrial dysfunctions arise from the early stages of SMA pathogenesis and may play a pivotal role in the disease’s progression, even by impacting directly on the level of SMN protein. Increased mitochondrial ROS production can affect *SMN2* splicing by reducing functional SMN levels. The female sex hormone estrogen (E2) regulates the expression of the transcription factors PGC-1α, NRF 1, and NRF2, which are important for the expression of the mitochondrial *TFAM*. Females seem to have better motor functions in a SMA mouse model, and the mouse brain showed lower oxidative stress in the mitochondria compared to males [109,110]. However, more information is needed to clarify this gender gap in SMA patients.

Finally, X-linked genes escape inactivation by creating a dosage imbalance between the two sexes. The genes *PLS3*, *USP9X*, and *UBA1* are located on the X chromosome and modulate the severity of SMA. Indeed, *USP9X* and *UBA1* expression are female-biased due to Xi escape, and females would also be variable with regards to PLS1 [36]. Other X-linked genes may have sex-specific effects on the phenotype of SMA, including the androgen receptor gene, cell-signaling receptor genes, and microRNA genes. In addition, some X-linked genes are associated with enhanced female immunity against infections and with sexual dimorphism of the immune inflammatory response [36]. Around 10% of human microRNA genes are located on the X chromosome and could potentially have sex-specific effects on RNA metabolism. Although some miRNAs have been reported to be possibly implicated in SMA pathogenesis among different SMA models and human samples, there are no publications regarding sex-linked differential expression. However, it is tempting to hypothesize that the sexual dimorphism in SMA could be partly regulated by X-linked miRNAs such as miR-92a-2-5p, the availability of which is predicted to be modulated by SMN circular RNAs [36].

## 7. Discussion

Gender differences in NDs have been frequently discussed. Starting from the incidence/prevalence ratio, it became evident that each disorder shows peculiarities by sex in most of its features, i.e., in pathophysiology as well as clinical outcomes. Investigating these sometimes-evident discrepancies can represent not only a valuable support in the study of NDs but, in our view, it may offer a solid background for innovative therapeutic approaches, which is of significant importance considering that, at present, most of these conditions still claim to require definitive treatments.

NDs are indeed a heterogeneous group of disorders affecting both the CNS and immune system. As summarized in Table 1, in the CNS, the main actors of innate immunity are the glial cells, mainly microglia, then astrocytes; peripheral cells (e.g., T-cells) are also involved, as well as circulating mediators and modulators like cytokines. Microglia are the resident macrophages within the CNS; they are dynamic cells that, through their connections, monitor the cerebral parenchyma, contribute to the maintenance of tissue homeostasis, and therefore perform a protective function, behaving like a “latent bystander” against any insults such as toxins, trauma, and injury [111].

Several observations reported that the microenvironment, interactions with other cells, and age are the main factors that trigger microglial activation [112]. Like all cells, microglia in fact also age, and therefore they are subjected to selective age-dependent changes. Aging is one of the major risk factors for NDs, and with a growing elderly population, their prevalence seems to increase; evidence also shows that the severity of most NDs also gets worse with age, as evidenced by impaired recovery following chronic insults [72].

Microglial cells are classified into two subpopulations: M1 cells are classically activated by LPS and IFNγ, express specific surface antigens, and produce high levels of oxidative metabolites, proteases, and proinflammatory cytokines such as IL-1β, TNFα and IL-6. In healthy conditions, they play a central defensive role against insults such as pathogens and tumor cells. In contrast, M2 cells are alternately activated; they show anti-inflammatory activity, induced by IL4-13, and express CD206 and arginase I, which downregulate inflammation by promoting tissue repair. Some theories suggest that an alteration of the M1/M2 ratio leads to the initiation of pathologies such as NDs, and those that are protective functions are transformed into damage for the cells and the host organism—hence the so-called microglia’s “Yin and Yang”, or “Vicious Cycle” [112,113]. In particular, it seems that chronic activation of M1 cells produces pro-inflammatory mediators such as TNFα and reactive oxygen species (ROS), hydroxyl radicals, and superoxide anions that negatively impact neurons, leading to the progressive neuronal loss that is the hallmark of neurodegeneration [114]. It is also worthy to note that some of these players change during the course of each disease, as e.g., the M1 phenotype shifts to M2 as the disease progresses, thus confirming their involvement not only in the onset of the disease (see Table 1). Sex differences in microglia number, morphology, immune molecule load, and transcriptomics exist starting from the early phases of neurodevelopment, as they are related to age, brain region, hormonal, and environmental factors. In animal models of neurological disorders, this seems to justify the differences between the two sexes, as each uses different mechanisms in order to achieve similar functional states [115]. As an example, microglia in the adult female brain are less responsive than those in the male brain, and inflammatory events in adulthood seem to promote age-related microglial dysfunction in females, which is correlated with worsened functional outcomes [116].

Preclinical and clinical studies reveal significant differences in male and female brains in both physiological and pathological conditions that cannot always be explained by the immunological path. Although sexual dimorphism might be the key to understanding the discrepancies in incidence, outcome, and progression of several NDs [115], the impact of sex/gender is still far from being completely elucidated. 

Many further factors may explain striking gender differences (Figure 2). Complicated networks of physiological dimorphism, genetics, epigenetics, hormonal profile, immune response, gut microbiota, environmental exposure, and lifestyle influence the degenerative manifestations of diseases such as MS, PD, AD, ALS, and SMA. However, since the complexities involved are multifactorial and influenced by the course of life and its associated changes in the physiological and anatomical premises, it remains a challenge to claim a precise intervention.

Finally, some mechanisms in the different pharmacokinetic response between males and females should also be considered, whereby men appear to have greater activity than women, e.g., for cytochrome P450 (CYP). Other important aspects are related to physiological differences, such as generally lower body weight and organ size, a higher percentage of body fat, a lower glomerular filtration rate, and different gastric motility in women compared to men [116]. The coexistence of gender-related autoimmune mechanisms may represent a therapeutic approach that requires further attention in NDs.

The need for personalized medicine according to sex and gender is now generally recognized, but to date no specific guidelines are yet available. More insights are also necessary with the aim of assessing how sex/gender may affect the safety, tolerability, and most importantly, the effectiveness of medications. We believe that, by looking at these differences, we can help address more targeted therapeutic interventions.

## Figures and Tables

**Figure 1 ijms-24-06354-f001:**
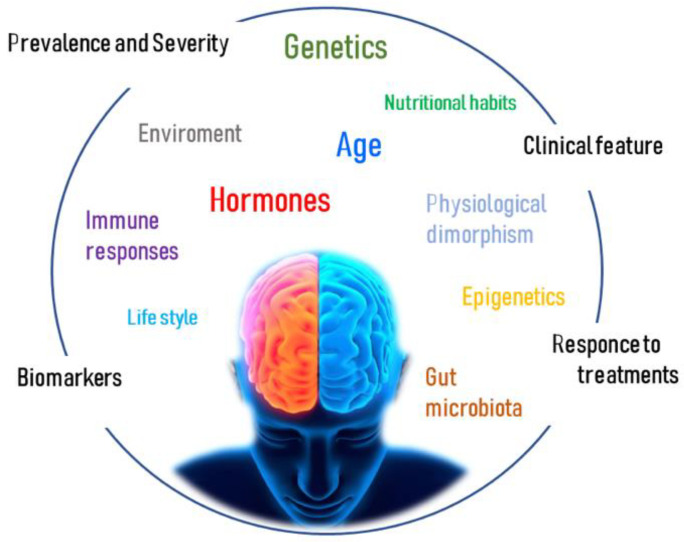
Multifactoriality of gender differences in neurodegenerative diseases. Factors including age, genetics, epigenetics, gonadal hormones, immune responses, physiological dimorphism, gut microbiota, lifestyle, nutritional habits, and environment may interact to determine gender differences, which result in variations of prevalence and severity, clinical features, biomarkers, and response to treatment between affected females and males.

**Figure 2 ijms-24-06354-f002:**
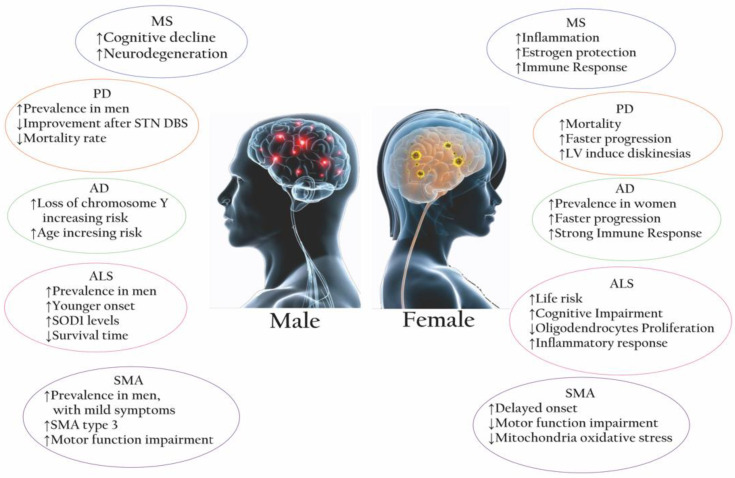
Summary of the main sex-related differences in neurodegenerative diseases. In pathologies like MS, PD, AD, ALS, and SMA, epidemiological, pathogenic, and clinical features show a distinctive pattern between affected females (**right side**) and males (**left side**). Abbreviation: MS = multiple sclerosis; PD = Parkinson’s disease; AD = Alzheimer’s disease; ALS = amyotrophic lateral sclerosis; SMA = spinal muscular atrophy; STN DBS = subthalamic nucleus deep brain stimulation; LV = levodopa; SOD1 = superoxide dismutase type 1.

**Table 1 ijms-24-06354-t001:** Summary of the immune/inflammatory features in the studied NDs. The first column indicates the diseases, in the second the female/male ratio; the central columns report the main differences—if any—of peripheral/central inflammatory cells/mediators between the two sexes, whereas in the last column on the right there are those in common, possibly related to the course of each disease.

NDs	Female/Male Ratio	Main IMMUNE FEATURES (In Vivo/Vitro)		
		Female	Male	In Common
MS	3:1 [10]	Higher neutrophils/macrophages activity [11] Higher CD4^+^ T cell, CD4^+^/CD8^+^ ratio [11] APCs are more competent [11] Higher PGR expression in microglia [17] Higher expression of IL-21, IL-27, and IL-18 [18] Notable Treg, TH1/TH2 variability [18,19]. (*Mice*) Higher Th1 cytokine production [20]	Higher NK cells [11] Higher CD8^+^ Tcell [11] Higher CD3^+^ and TNFα [21] Higher IL-1β and TNF [17] APCs secrete IL-10 [21] (*Mice*) Higher lymphocyte infiltration [20]	M1 in early MS shifts to M2 in later stages [22] Patients with more severe disease have higher proportions of lesions with foamy microglia/macrophages [17] TNFα is increased by macrophages/microglia during the early development of sclerotic plaques [21]
PD	1:1.5 [8]	Strong activation of peripheral monocytes. Consistent signature of changes in inflammatory signals (e.g., natural killer cell-mediated cytotoxicity pathway, APC, cytokine-cytokine receptor pathway) [23] (*Mice*): enhanced expression of IP-10 in astrocytes [24]	(*Mice*): enhanced expression of IL-1, IL-2, IL-6, and TNF-α in astrocytes [24]	The cytokine inflammatory signature and a-synuclein-specific T-cell reactivity are intense in the preclinical/early stage but may wane in more advanced PD [23]
AD	2:1 [25]	Increased inflammatory cytokines, chemokines, and gliosis [11,26] Activation of neurons in the active phase [27] (*Mice*): rod-shaped microglia with compromised phagocytic capacity [28] and an early state of alertness to inflammatory stimuli with accumulated Ab [29]	Activation of olygodendrocytes in the active phase (*Mice*): amoeboid microglia with greater phagocytic capacity [28]	Neuron-glia interactions: microglia adopt and activate an inflammatory phenotype by shifting to glycolysis [28]
ALS	1:1.2–1.5 [30]	Stronger immune response (STAT3 activation) [31] Higher expression of inflammatory genes in macrophages and innate immune cells [17]		Increased T cells, macrophages, IL-17A [32], IL-1β, TNFα, ROS, and NO [33] Lower levels of CD4^+^/higher levels of CD8^+^ [33] Release of proinflammatory cytokines by TLR2, TLR4, CD14 [34] Higher expression of IL2, IL5, IL6, IL8 [31], IL-13 [33] Higher levels of IL-7, IL−6, TNFa, TNF-R1, TNF-R2 [35] Microglia shift from the M1 to the M2 phenotype as soon as the disease progresses [22]
SMA	1:2 [36]	(*Mouse*): lower oxidative stress in muscular mitochondria, milder involvement of inflammatory pathways [36]	(*Mouse*): higher male susceptibility to the cumulative effects of oxidative stress [36]	T-cell alterations to an abnormal neuroinflammatory response and disease exacerbation [37] Increased astrogliosis [20] Increased levels of proinflammatory cytokines and neurotrophic factors in the CSF of active SMA1 patients [38]

## Data Availability

Not applicable.

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
