# Peer review of "Sex and Gender Differences in Neurodegenerative Diseases: Challenges for Therapeutic Opportunities"

_ijms, 2023, doi:10.3390/ijms24076354_

Round 1
Reviewer 1 Report
In this article, the authors briefly and somewhat superficially review 5 neurodegenerative (ND)diseases, multiple sclerosis, Parkinson’s disease, Alzheimer’s disease, amyotrophic lateral sclerosis, and spinal muscular atrophy, highlighting sex/gender differences in their pathology. Overall, the manuscript is well written and is potential interest to students who may want a general introduction to the subject matter. The manuscript would benefit by including age as a potential variable. So, sex, gender, and age differences in ND. If there are any gender differences in responses to standard of care, those should also be addressed.
Author Response
Reviewer-1:
In this article, the authors briefly and somewhat superficially - review 5 neurodegenerative (ND)diseases, multiple sclerosis, Parkinson’s disease, Alzheimer’s disease, amyotrophic lateral sclerosis, and spinal muscular atrophy, highlighting sex/gender differences in their pathology.
Overall, the manuscript is well written and is potential interest to students who may want a general introduction to the subject matter.
The manuscript would benefit by including age as a potential variable. So, sex, gender, and age differences in ND.
We really appreciate the suggestion from the reviewer; however, we respectfully decline to add another potential variable since the main the scope of this review was to shed more lights in the sex/gender differences in NDs by looking at several possible contributors.
However, we agree that age is of significant importance in NDs as in sex differences. As a matter of fact, we respectfully would like to point out that the impact of age in pathophysiology as well as in clinical features like the response to treatment has been already reported in several sections of this manuscript (see: line 155 for MS, line 167 in PD, line 301, 308, 324, 344 in AD, line 391, 407-408 in ALS).
Furthermore, also following the reviewer’s suggestion, we briefly addressed this issue by adding some notes in the discussion (lines 530-535). We also added AGE in the multifactorial representation of FIGURE-1.
Please also consider that, in order to summarize some the examined issues (in terms of sex difference from the immunological point of view) and to possibly address this reviewer’s concern about the accuracy of this review, we added a Table, and we made substantial changes in the Discussion.
If there are any gender differences in responses to standard of care, those should also be addressed.
We apologize if this issue was not clearly addressed; we made some changes throughout the manuscript to better also explain this important subject.
In particular, please look at the following reference for each disease:
- MS: lines 150-161
- PD: lines 231-265
- AD: lines 352-263
- ALS: lines 429-435 (no data available on the issue)
- SMA: 451-453 (no data)
Reviewer 2 Report
This manuscript is mainly factual, and the title is more attractive than the main text.
Many generalities are often stated (for ex page 5 lines 198 to 203) oxidative stress and neuroinflammation: which mediators are involved in the PD?
The figures do not provide sexual specific differences between man and woman.
The authors did not discuss the substantial relationship with a specific increase in one or several inflammatory response(s) according to the sex, and thus to the development of the disease.
It could have been interesting to produce a Table about the presence/absence of a sexual difference belonging to the innate immunity and/or activation of inflammation after the beginning of a neurodegenerative disease.
Also, to ask about inflammatory actors in men and women; Are these mediators different between the two genders all along the disease, or may have a different time-course?
What are the main inflammatory cells (microglia, astrocytes...) implied in the differences observed between M and W?
Author Response
Reviewer-2:
This manuscript is mainly factual and the title is more attractive than the main text.
We regret to realize that this review did not reach this reviewer’s expectation. In our view, we wanted to address the topic by giving cause for reflection, starting from evidence in the different diseases. Anyway, as detailed below, we made substantial changes in the overall manuscript, hopefully that this will improve the overall value of the review.
We also believed that the text reflected the overall message from the review. As clearly highlighted in the revised version of this manuscript, we made substantial changes also following her/his helpful suggestions. We also made a slight but significant change in the title; however, we are willing to make further adjustments in both the sections.
Many generalities are often stated (for ex page 5 lines 198 to 203) oxidative stress and neuroinflammation: which mediators are involved in the PD?
We thank the reviewer for this targeted comment; this section has been improved with more information, as you can find from lines 166 to 172, and more extensively in the interval of lines 212-230.
The figures do not provide sexual specific differences between man and woman.
We have improved the legend of Figure-1 and the draft of Figure-2; for the latter, we would like to pinpoint that all the differences between the two sexes have been specified from the left to the right sight of the picture. However, we are willing to make further changes, according to this reviewer’s suggestion.
The authors did not discuss the substantial relationship with a specific increase in one or several inflammatory response(s) according to the sex, and thus to the development of the disease.
As better detailed in the following reply, we made some changes in the discussion and in the new added Table in order to possibly answer to this issue.
It could have been interesting to produce a Table about the presence/absence of a sexual difference belonging to the innate immunity and/or activation of inflammation after the beginning of a neurodegenerative disease.
Also, to ask about inflammatory actors in men and women; Are these mediators different between the two genders all along the disease, or may have a different time-course?
What are the main inflammatory cells (microglia, astrocytes...) implied in the differences observed between M and W?
We really thank the reviewer for these very helpful comments; as she/he can find now, we have added a Table with most of the requested information, in particular about the “immune actors” involved in the analyzed NDs, by sex, and in common between them. Furthermore, in some cases we have found that there are time-course changes, as indicated in the last column of the table. We have also recalled this concept in the revised version of the discussion – see lines 550-553.
Overall, we made substantial changes in the discussion by focusing the immunological implications and the main cellular/molecular players that seem to be involved in the onset and course of the studied diseases, and in general in the neurodegeneration processes.
We understand that we did not address all the issues, but we believe we have offered some interesting subjects for the discussion.
Round 2
Reviewer 1 Report
The authors have adequately addressed my initial concerns.
Reviewer 2 Report
I thank the authors for their revision, and Table 1 (in the present form) becomes a very interesting tool for researchers. A lot of molecules/mechanisms are still missing in our knowledge or sexual differences between male and female. I hope that such a review may promote further scientific researches by using both sexes (in vivo as well in vitro).